# Effects on Health-Related Quality of Life of Biofeedback Physiotherapy of the Pelvic Floor as an Adjunctive Treatment Following Surgical Repair of Cystocele

**DOI:** 10.3390/jcm9103310

**Published:** 2020-10-15

**Authors:** Pedro-Santiago Borrego-Jimenez, Bárbara-Yolanda Padilla-Fernandez, Sebastián Valverde-Martinez, Maria-Helena Garcia-Sanchez, Maria-de-la-O Rodriguez-Martin, Maria-Pilar Sanchez-Conde, Maria-Carmen Flores-Fraile, Magaly Marquez-Sanchez, Javier Flores-Fraile, Miguel Perán-Teruel, José-Antonio Mirón-Canelo, Maria-Begoña García-Cenador, María-Fernanda Lorenzo-Gómez

**Affiliations:** 1Department of Surgery, University of Salamanca, 37001 Salamanca, Spain; pedro.borrego@gmail.com (P.-S.B.-J.); sebasv_2000@hotmail.com (S.V.-M.); pconde@usal.es (M.-P.S.-C.); maria.flores.fraile@usal.es (M.-C.F.-F.); mbgc@usal.es (M.-B.G.-C.); mflorenzogo@yahoo.es (M.-F.L.-G.); 2Physiotherapy Department of Institute of Applied Technology, Abu Dhabi 3798, UAE; 3Section of Urology, Department of Surgery, University La Laguna, 38200 San Cristóbal de la Laguna, (Santa Cruz de Tenerife), Spain; padillaf83@hotmail.com; 4Multidisciplinary Renal Research Group of the Institute for Biomedical Research of Salamanca (IBSAL), 37007 Salamanca, Spain; magalymarquez77@gmail.com (M.M.-S.); miroxx@usal.es (J.-A.M.-C.); 5Department of Urology of University Hospital of Avila, 05071 Ávila, Spain; 6Department of Gynecology and Obstetrics of University Hospital of Salamanca, 37007 Salamanca, Spain; helenagmarter@hotmail.com (M.-H.G.-S.); mrodriguezm@yahoo.es (M.-d.-l.-OR.-M.); 7Department of Anesthesiology of University Hospital of Salamanca, 37007 Salamanca, Spain; 8Department of Urology of Arnau de Vilanova Hospital, 46015 Valencia, Spain; mperanurologo@gmail.com; 9Department of Biomedical and Diagnostic Sciences of University of Salamanca, 37007 Salamanca, Spain; 10Department of Urology of University Hospital of Salamanca, 37007 Salamanca, Spain

**Keywords:** quality of life, cystocele, biofeedback, physiotherapy

## Abstract

Objectives: to demonstrate the benefits of physiotherapy (PT) with pelvic floor biofeedback (BFB) in improving health-related quality of life when used as a complementary therapy after surgical treatment of cystocele, in cases in which perineal pain or discomfort persists. Materials and methods: prospective observational study in 226 women who received complementary therapy after surgical treatment of cystocele due to persistent perineal discomfort or pain. Groups: GA (*n* = 78): women treated with 25 mg of oral pregabalin every 12 h plus BFB, consisting of 20 once-weekly therapy sessions, each 20 min long, with perineal pregelled surface electrodes connected to a screen which provides visual feedback; GB (*n* = 148): women treated with oral pregabalin 25 mg every 12 h without BFB. Variables: age, body mass index (BMI), time since onset of cystocele prior to surgery (TO), SF-36 health-related quality of life survey score, diseases and concomitant health conditions, follow-up time, success, or failure of postsurgical treatment. Results: average age 67.88 years (SD 12.33, 30–88), with no difference between GA and GB. Average body mass index (BMI) 27.08 (SD 0.45, 18.74–46.22), with no difference between GA and GB. Time since onset of cystocele prior to surgery (TO) was 6.61 years (SD 0.6), with no difference between GA and GB. Pretreatment SF-36 score was lower in GA success than GB success. Treatment was successful in 141 (63.20%) women and failed in 82 (36.80%). PT and age were the main predictors of success, and the least important were pretreatment SF-36 and the time elapsed after the intervention. In GA, 63 women (80.80%) showed improvement while 15 (19.20%) did not. Age was the main predictor of treatment success, while the least important was BMI. In GB, 78 women (53.80%) showed improvement while 67 (46.20%) did not improve. The main predictor was time since cystocele onset prior to surgery, while the least important was age. The odds ratio (OR) of improving quality of life for each unit increase in SF-36 was 11.5% (OR = 0.115) in all patients, with no difference between success and failure; in GA it was 23.80% (OR = 0.238), with a difference between success and failure; in GB it was 11.11% (OR = 0.111), with no difference between success and failure. GA and GB success had more history of eutocic delivery. GA success had more rUTI. GB success and GA failure both had more history of UI corrective surgery. The “failure” outcome had a higher number of patients with more than two concomitant pathological conditions. Conclusions: BFB as an adjunctive treatment improves quality of life in women suffering from persistent discomfort after surgery for cystocele. Young women who meet the criteria for recurrent urinary tract infection or who have a history of eutocic delivery show greater improvement. Body mass index does not influence response to treatment, while the presence of more than two concomitant conditions indicates a poor prognosis for improving quality of life.

## 1. Introduction

Bladder prolapse or cystocele is the most common type of prolapse, with an 11% probability of successful surgical repair, and symptoms of lumbar strain, vaginal pain, and mucosal lesions in many cases [1]. The cystocele is a frequent affectation in women from 50 years of age. Depending on the grade and whether or not it is accompanied by urinary incontinence, the therapeutic options considered are pelvic floor physiotherapy in grades 1–2, corrective surgery in grades 3–4. The indication for treatment, in any degree of cystocele, is mainly the symptoms, rather than the signs, that the patient refers to, which cause deterioration in the quality of life, rather than life risk. For this reason, the success of the treatment must be measured in the gain in quality of life related to health [2]. Surgical repair of cystocele is effective and safe with current procedures, although functional recovery may vary from one person to another [3,4,5]. Following surgical repair of the cystocele, there may be abnormalities in the healing process which may be related to the structural imbalance caused by the cystocele itself and could, in addition, lead to its recurrence [6].

Biofeedback or biological feedback is defined as the reintroduction into a biological system of the data obtained through its phenomenological study in such a way that the system modifies its response. It is a form of learning through a closed feedback “loop”, where one or more of the patient’s normally unconscious physiological processes are made evident by a visual, auditory and/or tactile signal [7]. Pelvic floor biofeedback (BFB) is a fundamental and indispensable technique in perineo-sphincter reeducation; it gives more immediate feedback than the therapist’s hand, and permits faster learning and awareness of perineal effort, both contraction and relaxation. Electromyographic biofeedback (EMG-BFB) has been shown to be effective, without side effects or complications, since 1992 [8].

This study was conducted on women who, after undergoing surgery for cystocele, continued to experience perineal pain or discomfort postintervention, in no case de novo, since the main indication for surgical repair of cystocele was precisely the existence of cystocele-related pain or discomfort. The objective was to demonstrate the benefits of physiotherapy (PT) with pelvic floor biofeedback (BFB) in improving health-related quality of life when used as a complementary therapy after surgical treatment of cystocele in cases in which perineal pain or discomfort persists.

## 2. Methods

International prospective observational study: out of a total of 1125 patients treated surgically by practitioners in the multidisciplinary research group, this study selected those women who required specialized management in the mediate and late postoperative period (more than one month after surgery) due to persistent discomfort or perineal pain after surgical correction of the cystocele. Before starting the study, all symptoms, urological and extra-urological, were evaluated.

Only women whose symptoms were related only to the cystocele, that is, the prolapse and the sensation of weight, discomfort, even perineal pain caused by the cystocele, were included in this study. Women whose main symptom was urinary incontinence, frequency or urgency were not included.

Thus, a total of 226 women with perineal pain or discomfort after cystocele surgical treatment (discomfort or pain after cystocele surgery: DPACS) were included.

### 2.1. Sample Selection

Inclusion criteria: all women older than 18 years with DPACS between December 2013 and December 2016 were included in the study in a sequential, successive, exhaustive, prospective manner. A single surgical technique (modified Shull’s technique [9]) was used for all patients in our series.

Exclusion criteria: patients who had undergone surgery for cystocele and did not present with pain or perineal discomfort after surgical correction of the cystocele, or where discomfort was due to urinary tract or genital infection, urolithiasis, genitourinary congenital anatomical abnormalities, neurogenic bladder, intermittent catheterization, indwelling urinary catheter; pregnancy.

### 2.2. Study Groups

GA (*n* = 78): DPACS patients given oral pregabalin (25 mg) every 12 h plus adjuvant BFB.

GB (*n* = 148): DPACS patients given oral pregabalin (25 mg) every 12 h with no biofeedback.

All the women underwent transvaginal repair of cystocele. Surgical intervention involves an inflammatory reaction caused by the surgery itself, with or without implantation of prosthetic material, and only side effects and chronic discomfort which last beyond the normal postoperative period have been taken into account [10]. All patients were referred by protocol to the Pain Unit, where pregabalin with or without BFB was prescribed.

### 2.3. Procedure

The EMG-BFB therapeutic program consisted of therapy sessions during which the patient controls a signal on a screen by employing or working the appropriate perineal muscles. The patient is instructed to perform the procedure once a week, in a 20-min session, for a period of 20 weeks. The patient is placed in the supine position, with slight hip flexion and protection of the lumbar lordosis to avoid fatigue. In this position, the patient must be able to view the screen of the EMG-BFB equipment. Pregelled pediatric self-adhesive electrodes are used.

After a brief explanation of the anatomy of the pelvic floor, the patient was instructed to contract the perineal muscles for 3–5 s and relax for 6–8 s. These contractions were recorded, reflecting muscle tone and power as well as the duration of the entire perineal recording. Each signal was continuously recorded on a polygraph.

### 2.4. Variables

Age, body mass index (BMI), time since onset of cystocele prior to surgery (TO), SF-36 health-related quality of life survey score, diseases and concomitant health conditions, follow-up time, success, or failure of postsurgical treatment.

In the analysis of responses to the SF-36 QoL survey, Spanish version [11], two values were used:

“Pretreatment SF-36”: responses recorded at the time of diagnosis and prescription of surgical treatment for cystocele

“Post-treatment SF-36”: mean result of the responses recorded in follow-up examinations at 3 and 6 months after surgical correction of the cystocele, in all patients, both those who received pelvic floor biofeedback and those who received oral gabapentin.

Treatment outcome: treatment **success** was defined as either:

(a) SF-36 HRQoL total score equal to or greater than 80 points.

(b) SF-36 HRQoL total score 30 or more points higher than the initial situation.

The following two outcomes were considered treatment **failure**:

(a) SF-36 HRQoL total score below 80 points.

(b) SF-36 HRQoL increased by fewer than 30 points with respect to the initial situation.

### 2.5. Statistical Analysis

Results were analyzed with descriptive statistics, Student’s t, Chi2, Fisher’s exact test, ANOVA (with Scheffe’s test for normal samples and Kruskal−Wallis for other distributions), multivariate analysis and logistic regression, *p* < 0.05 was considered statistically significant.

### 2.6. Ethical Concerns

The study protocol with Code 230/284/1 was approved by the Ethical Committee for Clinical Research of the Ávila University Healthcare Complex (*Comité Ético de Investigación con Medicamentos de Complejo Asistencial Universitario de Ávila*).

## 3. Results

Three patients from GB were lost to follow-up. Mean age was 67.88 years (SD 12.33, median 67, range 30–88). There was no difference between GA (mean 60.92 years, SD 1.39, median 65, range 30–76) and GB (mean 66.84 years, SD 1.27, median 69, range 30–88) (*p* = 0.1024). Mean body mass index (BMI) was 27.08 (SD 0.45, median 26.68, range 18.74–46.22). There was no difference between GA (mean 26.19, SD 0.39, median 26.17, range 21.78–32.46) and GB (mean 27.97, SD 0.50, median 27.18, range 18.74–46.22) (*p* = 0.3275).

Mean time since cystocele onset (TO) prior to start of treatment was 6.61 years (SD 0.6, median 5, range 0.1941–30). There were no differences between GA (mean 6.33 years, SD 0.61, median 5, range 1–13) and GB (mean 6.88 years, SD 0.59, median 5, range 0.1941–30) (*p* = 0.2997).

Pretreatment SF-36 score was lower in GA success (mean 57.00, SD 2.115, range 54–61) than in GB success (mean 60.00, SD 3.16, range 54–65) (*p* = 0.0130).

There was no difference in pretreatment SF-36 between GA failure (mean 55.00, SD 0.516, range 55–56) and GB failure (mean 61.00, SD 3.232, range 52–63) (*p* = 0.1065).

There was no difference in post-treatment SF-36 between GA success (mean 84.00, SD 2.158, range 80–87) and GB success (mean 84.00, SD 3.05, range 80–89) (*p* = 0.3971).

Post-treatment SF-36 in GA failure (mean 56.00, SD 1.033, range 56–58) was lower than that of GB failure (mean 61, SD 2.381, range 55–63) (*p* = 0.0195).

A multivariate two-step cluster analysis was performed to identify differences, in the general sample and the groups, between patients whose treatment succeeded or failed (Table 1). Overall, treatment was successful in 141 women (63.20%), while HRQoL did not improve in 82 (36.80%). The main predictors of success in the general sample were PT and age, and the least important were pretreatment SF-36 score and time elapsed since the intervention (Figure 1).

In GA, treatment was successful in 63 women (80.80%) and 15 (19.20%) did not improve. Age was the main predictor of success, while the least important was BMI (Figure 2).

In GB, treatment was successful in 78 women (53.80%), while 67 (46.20%) did not improve. The main predictor of treatment success was time since onset of cystocele prior to surgical repair, while the least important was age (Figure 3).

A logistic regression model with Nagelkerke R-Squared 1.0000 goodness-of-fit was used. The odds ratio for improved QoL was calculated: for each unit increase in the post-treatment SF-36 score, quality of life increased by 11.5% (OR = 0.115) in the general group, with no difference between success and failure (*p* = 0.9930). In GA, the probability of improving HRQoL was 23.80% (OR = 0.238) for each unit increase in SF-36, with a marked difference between success and failure (*p* = 0.0001). In GB, the probability of improving HRQoL was 11.11% (OR = 0.111) for each unit increase in SF-36, with no difference found between success and failure (*p* = 0.994) (Table 2).

Table 3 and Table 4 show the distribution of the concomitant diseases and conditions in GA and GB, success, and failure, respectively. Urinary tract infection, overactive bladder, or de novo urinary incontinence recorded in Table 3 and Table 4, which had been considered exclusion criteria, appeared after cystocele intervention.

## 4. Discussion

In our multicenter, multidisciplinary research team, with a high volume of cystocele treatments, both with conservative and surgical measures, we have investigated from the medical factors the condition complications of the intervention, intervention modalities that give better results [3,4,5,12,13,14,15,16,17,18,19,20,21,22,23,24], and now, in this study, we investigate the benefits from treatment with pelvic floor biofeedback on discomfort caused by cystocele when said discomfort is not corrected with surgical treatment. The annual incidence of hospital admission with a diagnosed prolapse is 0.204%, while the corresponding incidence of prolapse repair surgery is 0.162%. An unknown number of women with pelvic organ prolapse are attended without hospitalization or surgery, while others never seek medical attention; the incidence and prevalence of this condition is therefore believed to be an underestimate. No published data on spontaneous remission exist [25].

Although ages did not differ between GA and GB (*p* = 0.1024), the age range was very wide in both groups (30 to 88 years). The main predictors of success in the general sample were the time since cystocele onset and patient age. We note that age is the main predictor of success in women treated with BFB, while BMI has the least impact. This finding is highly relevant when prescribing BFB in women with cystocele-related discomfort following a surgery which did not correct all related symptoms, and even more so with the addition that elevated BMI is not a predictor of poor prognosis.

Time since cystocele onset (TO) is important across all women, but still more so in those who will receive only pharmacological treatment (pregabalin in our case). Urging women to seek early treatment for their cystocele-related perineal symptoms is therefore of the greatest importance, as TO was observed to be a major prognostic factor: the sooner the cystocele is treated, the better the outcome of treatment.

Vaginal childbirth can cause muscle and nerve injuries and tissue rupture, among other problems, and traumatic deliveries are considered a risk factor for anterior compartment prolapse [25]. In our series, a history of eutocic delivery was more frequent in women for whom treatment was successful (19.04% in GA success and 30.86% in GB success, *p* = 0.1263) than in those who did not improve (0% in GA failure, 19.40% in GB failure). This finding is in line with other results from our research group, which found that a history of eutocic delivery is a predictor of a favorable outcome of pelvic floor treatments [26].

Women who met the criteria for RUTI according to the European Association of Urology guidelines [27] presented disparate results depending on whether treatment outcome was successful: 25.39% success rate in patients who received BFB, compared to 7.40% success in those who did not (*p* = 0.0044). On the other hand, patients with RUTI accounted for 33.33% of failures in GA and 17.91% in GB (*p* = 0.2874). These data support the benefit of BFB in improving voiding dynamics and control of UTIs, an important line in our research group [28,29].

In recent years, nonabsorbable mesh has been used in pelvic prolapse surgery. This technique is associated with various complications in the vagina, bladder, and intestine [30]. A subset of these patients develops chronic pain, possibly due to muscle and nerve irritation which may be caused by mesh implantation [31]. Transvaginal mesh surgery is the most frequent, employed in 80% of cases according to the FDA [32]. The vaginal route also allows the repair of cystocele with prolapse of the other chambers. Surgical correction, with or without sub urethral tape, is advised if the symptoms are distressing or if the cystocele is high-grade. Surgical repair of cystocele using anterior colporrhaphy with Kelly plication could fail if there are paravaginal defects, and a long-term recurrence of 15% has been reported when using Kelly’s technique. Figures for hidden UI after cystocele repair vary between 16–67% [33]. The present study used the modified Shull technique, especially suitable for correcting central and lateral cystoceles. Several sutures (4 or 6) are applied to the tendinous arch from the ischial spine upwards and are transferred to the pubocervical fascia and extramucosal vaginal wall [9]. In our experience, relapses are less than 10% (14). The prosthetic material implanted during surgery may affect results [24]. A single surgical technique (modified Shull’s technique [9]) was used for all patients in our series. However, in the 1125 total patients treated surgically by the practitioners in the multidisciplinary research group, 4 types of surgical repair were used: mesh for cystocele correction plus transobturator suburethral tape, TOT (chosen from the following: Gynecare^®^ (Ethicon Johnson-Johnson US LLC, Cincinnati, OH, USA), Monarc^®^ (AMS, Minnetonka, MN, USA), SAFYRE^®^ (Promedon, Córdoba, Argentina), Contasure KIM^®^ (Neomedic International), I-Stop^®^ (CL Medical, Massachusetts, MA, USA), DynaMesh^®^ (FEG Textiltechnik mbH, Aachen, Germany), Aris^®^ (Porgès-Coloplast, France), and Swing-Band^®^ (Balmer-Médical SA, Concise, Switzerland); mesh placement for cystocele correction, without use of TOT; surgical cystocele correction without mesh but with TOT; and surgical correction of the cystocele without mesh and without TOT. To avoid bias in the comparison, only the 226 women who underwent cystocele surgery with the modified Shull technique were chosen for this study, of whom only 20 in GA and 35 in GB had a Kim System transobturator suburethral tape. No biomaterial was used in the remaining patients. Our work group has researched, in depth, the repercussions of the use of various biomaterials on the pelvic floor [5,34,35,36,37] and our preference is to repair bladder prolapse via the vagina without the use of a cystocele mesh, using the modified Shull technique and placing a Kim System transobturator suburethral tape. No complications related to the suburethral tape were found for any case in this study.

Women who do not receive BFB but improve with pregabalin treatment alone frequently have a history of corrective surgery for UI: 60.49% in GB success versus 17.46% in GA success (*p* = 0.0001). Meanwhile, the same history is found in 66.66% of GA failure, but 32.83% of GB failure (*p* = 0.0204). This correlation is highly relevant given the intrinsic benefit of BFB for women suffering from UI [30]. In women who have already undergone UI surgery, have a cystocele and undergo surgery only for cystocele repair, any possible benefit of BFB for voiding dynamics is lost. Women who have not previously undergone surgery for UI, however, will benefit more from BFB.

Diabetes, vascular insufficiency, and congestive heart failure are considered decompensation factors [25]. No history of DM was found in the group treated with BFB, both successful and unsuccessful. Incidence of DM in GB, without BFB, was slightly higher in cases of treatment failure (7.46%) than in treatment success (4.93%).

On the other hand, women in poorer health, defined as the presence of more than two concomitant pathological conditions, were more frequent in GB success (35.80%) than in GA success (11.11%) (*p* = 0.0008). When treatment did not improve quality of life, patients in GA were in poorer health: more than two concomitant medical conditions in GA failure in 66.66% versus 52.23% in GB failure, with no significance (*p* = 0.3949). Overall, women with more conditions were in the failure group, both in GA and GB. However, concomitant diseases and health conditions could not be included in the multivariate analysis due to the low number of cases, and only univariate analysis was performed.

The SF-36 questionnaire is a generic measure of HRQoL, used internationally and validated in Spain [11]. We used it in the field of urology and, specifically, in pelvic floor interventions. Group GB, without treatment with BFB, featured the highest pretreatment SF-36 HRQoL scores. In GA patients, however, BFB makes an important contribution. Cystocele surgery is indicated when the cystocele causes multiple symptoms or when it is a high-grade cystocele, grade 3 or grade 4, with multiple associated inflammatory pathologies, such as recurrent infections (due to postvoid residual urine) and pain connected to the distension of the pelvic floor when the bladder falls through the vagina. BFB therefore provides a high success rate as an adjunct to surgical treatment by focusing its effects on stabilizing and reinforcing the pelvic floor, reducing painful symptoms associated with the structural alteration of the pelvic floor in these women, should specific treatment be required for significant, long-term (more than one month after surgery) discomfort or pain.

## 5. Conclusions

When used as an adjunctive therapy, pelvic floor biofeedback improves quality of life in women suffering from persistent discomfort after surgery for cystocele. Young women who meet the criteria for recurrent urinary tract infection or have a history of eutocic delivery show greater improvement. Body mass index does not influence the response to treatment, while the presence of more than two concomitant conditions indicates a poor prognosis for improving quality of life.

## Figures and Tables

**Figure 1 jcm-09-03310-f001:**
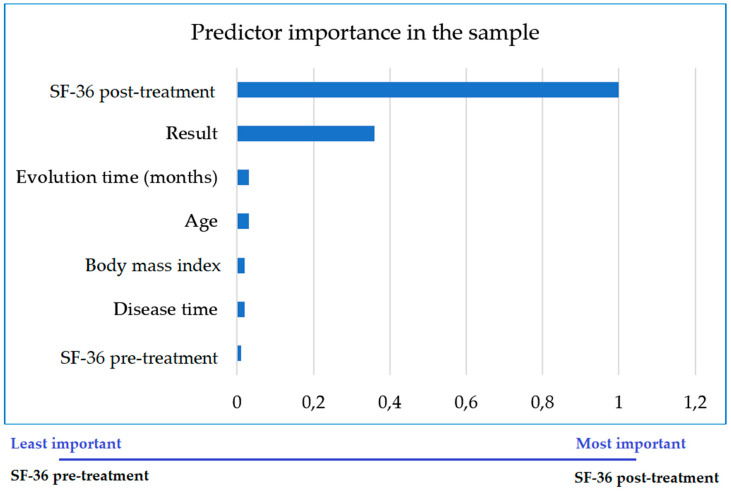
Importance of each variable as a predictor of the successful outcome of treatment in improving quality of life in all patients.

**Figure 2 jcm-09-03310-f002:**
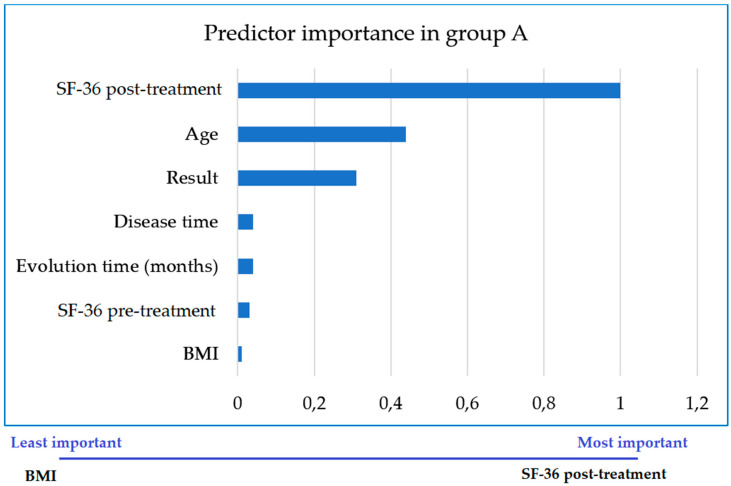
Importance of each variable as a predictor of the successful outcome of treatment in improving quality of life in women treated with adjunctive biofeedback (GA). BMI-Body Mass Index.

**Figure 3 jcm-09-03310-f003:**
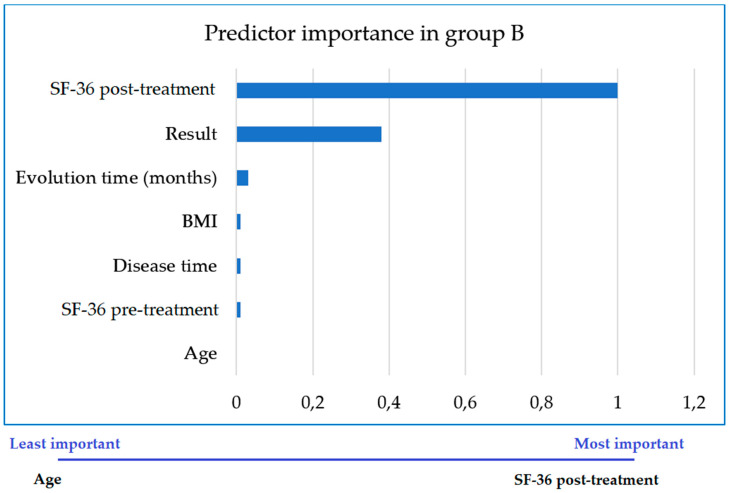
Importance of each variable as a predictor of the successful outcome of treatment in improving quality of life in women not treated with pelvic floor biofeedback (GB).

**Table 1 jcm-09-03310-t001:** Age, BMI, time since onset of cystocele, follow-up time, SF-36 before and after treatment.

**Group/Subgroup**	**Overall Sample: Success**			**Overall Sample: Failure**		
**Variable**	**Mean**	**Standard Deviation (SD)**	**Range**	**Mean**	**Standard Deviation (SD)**	**Range**
Age	68.00	8.525	40–88	65.00	16.52	30–85
BMI	26.67	5.20	20.86–46.22	27.53	4.53	18.74–41.91
Time Since Onset in Months	72.00	14.83	2.33–360.00	72.98	32.989	37.47–324.01
Follow-up Time	7.28	5.20	1.00–21.00	8.00	6.16	1.00–30.00
SF-36 Pretreatment	58.00	3.005	54–65	58.00	3.24	52–63
SF-36 Post-treatment	84.00	2.697	80–89	59.00	2.45	55–63
**Group/subgroup**	**GA: Success**			**GB: Failure**		
**Variable**	**Mean**	**Standard Deviation (SD)**	**Range**	**Mean**	**Standard Deviation (SD)**	**Range**
Age	67.00	4.823	60–76	30.00	9.812	30–49
BMI	24.99	4.12	21.77–32.46	32.46	5.51	21.77–32.46
Time Since Onset in Months	21.41	45.89	9.97–88.05	188.05	30.74	128.51–188.05
Follow-up Time	10.70	3.82	5.74–15.67	15.67	2.56	10.70–15.67
SF-36 Pretreatment	57.00	2.115	54–61	55.00	0.516	55–56
SF-36 Post-treatment	84.00	2.158	80–87	56.00	1.033	56–58
**Group/subgroup**	**GA: Success**			**GA: Failure**		
**Variable**	**Mean**	**Standard Deviation (SD)**	**Range**	**Mean**	**Standard Deviation (SD)**	**Range**
Age	68.50	10.593	40–88	67.00	12.779	31–85
BMI	26.86	5.89	20.86–46.22	27.34	4.28	18.75–41.91
Time Since Onset in Months	55.03	145.46	2.33–360.00	69.69	365.62	37.47–324.01
Follow-up Time	5.00	5.82	1.00–21.00	5.00	6.28	1.00–30.00
SF-36 Pretreatment	60.00	3.16	54–65	61.00	3.232	52–63
SF-36 Post-treatment	84.00	3.05	80–89	61.00	2.381	55–63

Time since onset: time from cystocele onset until surgery, measured in months. Follow-up time: time with discomfort or pain following surgery, measured in months. BMI: body mass index.

**Table 2 jcm-09-03310-t002:** Results of the logistic regression analysis in the SF-36 health-related quality of life survey following surgical repair of the cystocele.

Logistic Regression: Odds Ratio of the Increase in Post-Treatment SF-36 QoL						
	B	S.E.	Wald	df	Sig.	Exp (B)
General	−2.167	236.898	0.0008	1	0.993	0.115
GA	−1.435	0.287	24.951	1	0.0005	0.238
GB	−2.200	293.413	0.0005	1	0.994	0.111

Variable(s) entered in step 1: SF-36 post-treatment. B: odds ratio estimator; S.E.: standard error; Wald: measurement method; df: model; Sig.: Significance or *p*-value; Exp (B): odds ratio.

**Table 3 jcm-09-03310-t003:** Distribution of diseases and concomitant health conditions in GA and GB in which there was significant improvement in HRQoL SF-36 (success).

CONCOMITANT DISEASES OR CONDITIONS.	GA Success *n* = 63		GB Success *n* = 81		*p*
	*n*	%	*n*	%	
**Criteria For RUTI**	16	25.39	6	7.40	0.0044
Overactive Bladder	12	19.04	1	1.23	0.0002
Urinary Incontinence	7	11.11	0	0	0.0025
IC/Painful Bladder Syndrome	4	6.34	1	1.23	0.1682
Non-Operated Vesico-Urethral Reflux	0	0	1	1.23	1.0000
History of Corrective Surgery For UI	11	17.46	49	60.49	0.0001
History of Curettage	8	12.69	6	7.40	0.3963
History of Hysterectomy	0	0	9	11.11	0.0052
Other Surgical History	0	0	31	38.27	0.0001
History of Surgery for Gynecological Cancer	0	0	3	3.70	0.2566
History of Caesarean Section	0	0	2	2.46	0.5044
History of Surgery for Urethral Prolapse	0	0	2	2.46	0.5044
Botox	0	0	1	1.23	1.0000
CT With Benzodiazepines	7	11.11	17	20.98	0.1754
CT With Hypnotic (Zolpidem)	3	4.76	0	0	0.0815
CT With Anticholinergics	8	12.69	5	6.17	0.2422
CT With Topical Estrogens	7	11.11	3	3.70	0.1044
CT With Step 1 Analgesics	0	0	1	1.23	1.0000
CT With Step 2 Analgesics	0	0	3	3.70	0.2566
CT With Step 3 Analgesics	0	0	5	6.17	0.0679
CT With the Antidepressant Amitriptyline	0	0	16	19.75	0.0001
Treatment with Intravesical GAG Instillations	0	0	1	1.23	1.0000
Other Pharmacological Treatments	0	0	39	48.14	0.0001
Nulliparous	4	6.34	1	1.23	0.1682
History of Eutocic Childbirth	12	19.04	25	30.86	0.1263
History of Dystocic Childbirth	8	12.69	2	2.46	0.0215
Hypothyroidism	4	6.34	7	8.64	0.7560
DM	0	0	4	4.93	0.1315
HBP	0	0	19	23.45	0.0001
Depression	0	0	18	22.22	0.0001
Other Diseases or Pathological Conditions	0	0	43	53.08	0.0001
Smoker	4	6.34	5	6.17	1.0000
More Than 2 Concomitant Medical Conditions	7	11.11	29	35.80	0.0008
More Than 2 Past Surgical Interventions	8	12.69	30	37.03	0.0011
More Than 2 Concomitant Treatments	7	11.11	22	27.16	0.0211
Toxic Substance Use Plus Conc. Medical Conditions	0	0	5	6.17	0.0679
Toxic Substance Use Plus Conc. Surgical History	4	6.34	5	6.17	1.0000

RUTI: recurrent urinary tract infection. UI: urinary incontinence. IC: interstitial cystitis. CT: concomitant therapy. GAG: glycosaminoglycans. DM: diabetes mellitus. HBP: high blood pressure.

**Table 4 jcm-09-03310-t004:** Distribution of diseases and concomitant health conditions in GA and GB in which there was no significant improvement in HRQoL SF-36 (failure).

CONCOMITANT DISEASES OR CONDITIONS	GA Failure *n* = 15		GB Failure *n* = 67		*p*
	*n*	%	*n*	%	
**Criteria For RUTI**	5	33.33	12	17.91	0.2874
Overactive Bladder	5	33.33	10	14.92	0.1357
Urinary Incontinence	0	0	18	26.86	0.0333
IC/Painful Bladder Syndrome	0	0	1	1.49	1.0000
History of Corrective Surgery For UI	10	66.66	22	32.83	0.0204
History of Curettage	0	0	5	7.46	0.5786
History of Hysterectomy	5	33.33	9	13.43	0.1206
Other Surgical History	5	33.33	21	31.34	1.0000
History of Surgery for Gynecological Cancer	0	0	1	1.49	1.0000
History of Caesarean Section	0	0	2	2.98	1.0000
History of Surgery for Urethral Prolapse	0	0	1	1.49	1.0000
Botox	5	33.33	4	5.97	0.0087
CT With Benzodiazepines	0	0	8	11.94	0.3400
CT With Anticholinergics	0	0	5	7.46	0.5786
CT With Topical Estrogens	5	33.33	3	4.47	0.0044
CT With Step 1 Analgesics	5	33.33	2	2.98	0.0018
CT With Step 2 Analgesics	5	33.33	2	2.98	0.0018
CT With Step 3 Analgesics	5	33.33	4	5.97	0.0087
CT With the Antidepressant Amitriptyline	5	33.33	10	14.92	0.1357
Treatment with Intravesical GAG Instillations	0	0	2	2.98	1.0000
Other Pharmacological Treatments	0	0	25	37.31	0.0038
Nulliparous	5	33.33	1	1.49	0.0006
History of Eutocic Childbirth	0	0	13	19.40	0.1121
History of Dystocic Childbirth	0	0	1	1.49	1.0000
Hypothyroidism	5	33.33	2	2.98	0.0018
DM	0	0	5	7.46	0.5786
HBP	0	0	12	17.91	0.1107
Depression	0	0	6	8.95	0.5863
Other Diseases or Pathological Conditions	5	33.33	23	34.32	1.0000
Smoker	10	66.66	2	2.98	0.0001
More Than 2 Concomitant Medical Conditions	10	66.66	35	52.23	0.3949
More Than 2 Past Surgical Interventions	10	66.66	33	49.25	0.2631
More Than 2 Concomitant Treatments	5	33.33	18	26.86	0.7514
Toxic Substance Use Plus Conc. Medical Conditions	10	66.66	1	1.49	0.0001
Toxic Substance Use Plus Conc. Surgical History	10	66.66	1	1.49	0.0001

RUTI: recurrent urinary tract infection. UI: urinary incontinence. IC: Interstitial cystitis. CT: concomitant therapy. GAG: glycosaminoglycans. DM: diabetes mellitus. HBP: high blood pressure.

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
