# Peer review of "Effects on Health-Related Quality of Life of Biofeedback Physiotherapy of the Pelvic Floor as an Adjunctive Treatment Following Surgical Repair of Cystocele"

_jcm, 2020, doi:10.3390/jcm9103310_

Round 1
Reviewer 1 Report
Thank you for the manuscript and the topic. I am afraid, that a huge correction of the manuscript will be required. It will be essential for increasing the scientific quality of the article.
Minor comments.
Introduction:
the text might be replaced in 1 abstract.
Line 63: what do you mean, when writing “fibroblastic tissue”?
1.1. If both your sentences are from the one article, please, put the reference in brackets only after the second one.
1.2. The same as in the Introduction. Try to organize the text in another way – you can put all sentences in the one abstract.
2.8. You might put both sections in an appropriate place (they should be after the main body of the manuscript).
- Results
Please correct the fronts of the text in tables. Some legends should be improved. For me – the organization and way of results’ presenting are a little bit messy.
References: please correct the list of references due to the required standard.
Abbreviations: a list will be better than the table.
Major comments.
Please provide a deep background of the clinical problem, have studied by your group in the Introduction and Discussion sections.
The text should also have been corrected by a native speaker or agency.
Author Response
Thank you for the manuscript and the topic. I am afraid, that a huge correction of the manuscript will be required. It will be essential for increasing the scientific quality of the article.
Minor comments.
Introduction:
the text might be replaced in 1 abstract.
Answer: We have summarized the introduction to three paragraphs
Line 63: what do you mean, when writing “fibroblastic tissue”?
Answer:The phrase is confusing and is not essential, therefore we remove it.
1.1.-If both your sentences are from the one article, please, put the reference in brackets only after the second one.
Answer: We do it.
1.2.-The same as in the Introduction. Try to organize the text in another way – you can put all sentences in the one abstract.
Answer: We have summarized the introduction to three paragraphs
2.8. You might put both sections in an appropriate place (they should be after the main body of the manuscript).
Answer: We do it.
3.-Results
Please correct the fronts of the text in tables. Some legends should be improved. For me – the organization and way of results’ presenting are a little bit messy.
Answer: We have changed the legend of the figures and tables.
References: please correct the list of references due to the required standard.
Answer: We do it.
Abbreviations: a list will be better than the table.
Answer: We do it.
Major comments.
Please provide a deep background of the clinical problem, have studied by your group in the Introduction and Discussion sections.
We add this text in the INTRODUCTION:
The cystocele is a frequent affectation in the women from the 50 years of age. Depending on the grade and whether or not it is accompanied by urinary incontinence, the therapeutic options are considered: pelvic floor physiotherapy in grades 1-2, corrective surgery in grades 3-4. The indication for treatment, in any degree of cystocele, is mainly the symptoms, rather than the signs, that the patient refers to, which cause deterioration in the quality of life, rather than life risk. For this reason, the success of the treatment must be measured in the gain in quality of life related to health (1).
We add this text in the DISCUSSION:
In our multi-center multidisciplinary research team, with a high volume of cystocele treatments, both with conservative and surgical measures, we have investigated from the medical factors that condition complications of the intervention, intervention modalities that give better results (2-18), and now, in this study, we investigate the benefit from treatment with pelvic floor biofeedback to discomfort caused by cystocele when said discomfort is not corrected with surgical treatment.
The text should also have been corrected by a native speaker or agency.
ANSWER: We have review the text.
Reviewer 2 Report
This is an interesting paper on the effects on health-related quality of life of biofeedback physiotherapy of the pelvic floor as an adjunctive treatment following surgical repair of cystocele.
I have some remarks:
- line 104 Materials and methods: before starting the study, did you assess the presence of any symptoms other than perineal pain or discomfort (increased daytime frequency, urgency, incontinence, etc.)? If yes, please give the results. if not, please explain why.
- lines 115-117: You have listed among the exclusion criteria “patients...where discomfort was due to urinary tract or genital infection…”. But table 3 seems to show that 21 patients in group A (16 in the success group and 5 in the failure group) and 18 patients in the group B (6 in the success group and 12 in the failure group) were included in the study. Please explain this apparent inconsistency.
- line 125 statistical analysis: Please add the value of statistical significance.
- line 143: You wrote that you considered the answers to the SF-36 questionnaire recorded “at the time of diagnosis and prescription of surgical treatment for cystocele’ to evaluate the pre-treatment condition.
Did you also have the questionnaire filled in after the surgery and before starting treatment with gabapentin associated or not associated with BFB? If you have, please give them. If not explain why you decided to use the pre-op data.
- line 223: Please replace ‘CI: cistopatía interstiticial? With IC: interstitial Cystitis
- line 244: please replace (Bump and Norton 1998) with [14]
- line 272: you wrote. “A single surgical technique (modified Shull's technique [23]) was used for all patients in our series”. This is an inclusion criteria. Please, add in the paragraph ‘inclusion criteria’.
Author Response
This is an interesting paper on the effects on health-related quality of life of biofeedback physiotherapy of the pelvic floor as an adjunctive treatment following surgical repair of cystocele.
I have some remarks:
1.-line 104 Materials and methods: before starting the study, did you assess the presence of any symptoms other than perineal pain or discomfort (increased daytime frequency, urgency, incontinence, etc.)? If yes, please give the results. if not, please explain why.
ANSWER:
Before starting the study, all symptoms, urological and extraurological, were evaluated.
Only women whose symptoms were related only to the cystocele, that is, the prolapse and the sensation of weight, discomfort, even perineal pain caused by the cystocele, were included in this study. Women whose main symptom was urinary incontinence, frequency or urgency were not included.
These data are related to the following question from the same reviewer, where we explain that in Tables 3 and 4, urinary tract infection, overactive bladder, or de novo urinary incontinence recorded in Tables 3 and 4, which had been considered exclusion criteria, appeared after cystocele intervention, it means that appeared throughout the follow-up of the patients.
2- lines 115-117: You have listed among the exclusion criteria “patients...where discomfort was due to urinary tract or genital infection…”. But table 3 seems to show that 21 patients in group A (16 in the success group and 5 in the failure group) and 18 patients in the group B (6 in the success group and 12 in the failure group) were included in the study. Please explain this apparent inconsistency.
Indeed, it would be a contradiction between the exclusion criteria expressed in lines 115-117 and the results of diseases or concomitant health conditions in Tables 3 and 4. It happens that the exclusion criteria were that the patients suffered from urinary tract infection for example, at the time of being included in the study. Urinary tract infection, overactive bladder, or de novo urinary incontinence recorded in Tables 3 and 4, which had been considered exclusion criteria, appeared after cystocele intervention.
Therefore, we include this explanation in the results on lines 219-220.
- line 125 statistical analysis: Please add the value of statistical significance.
We add:
p <0.05 was considered statistically significant
- line 143: You wrote that you considered the answers to the SF-36 questionnaire recorded “at the time of diagnosis and prescription of surgical treatment for cystocele’ to evaluate the pre-treatment condition.
Did you also have the questionnaire filled in after the surgery and before starting treatment with gabapentin associated or not associated with BFB? If you have, please give them. If not explain why you decided to use the pre-op data.
We add:
in all patients, both those who received pelvic floor biofeeedback and those who received oral gabapentin.
- line 223: Please replace ‘CI: cistopatía interstiticial? With IC: interstitial Cystitis
We change it.
- line 244: please replace (Bump and Norton 1998) with [14]
We change it
- line 272: you wrote. “A single surgical technique (modified Shull's technique [23]) was used for all patients in our series”. This is an inclusion criteria. Please, add in the paragraph ‘inclusion criteria’.
We add it
Round 2
Reviewer 1 Report
Dear Authors,
Thank you for all corrections.
Generally, your text looks better after corrections. However, English is required further revision.
Reviewer 2 Report
Dear Authors,
this document, in my opinion, is now ready for publication. You had inserted new text and made modifications in answer to my comments and suggestions, and have satisfied all the points I made.